# Double-J Ureteral Stenting in Obstetrics and Gynecology: Pivotal or Problematic?

**DOI:** 10.3390/jcm13247649

**Published:** 2024-12-16

**Authors:** Viorel-Dragos Radu, Radu Cristian Costache, Pavel Onofrei, Pavel Banov, Feras Al Jaafari, Ingrid-Andrada Vasilache, Demetra Socolov, Rodica Radu

**Affiliations:** 1Department of Urology, Faculty of Medicine, University of Medicine and Pharmacy “Gr. T. Popa”, 700115 Iasi, Romania; viorel.radu@umfiasi.ro (V.-D.R.); radu.costache@umfiasi.ro (R.C.C.); 2Urological Department, “C.I. Parhon” University Hospital, 700115 Iasi, Romania; 3Department of Morpho-Functional Sciences II, Faculty of Medicine, University of Medicine and Pharmacy “Gr. T. Popa”, 700115 Iasi, Romania; 4Urological Department, Elytis Hope Hospital, 700010 Iasi, Romania; 5Department of Urology and Surgical Nephrology, “Nicolae Testemițanu” State University of Medicine and Pharmacy, MD-2004 Chisinau, Moldova; pavel.banov@usmf.md; 6Urology Department, Victoria Hospital, NHS Fife, Kirkcaldy KY2 5AH, Scotland, UK; feras.aljaafari2@nhs.scot; 7School of Medicine, University of St Andrews, St Andrews KY16 9AJ, Scotland, UK; 8Department of Mother and Child Care, University of Medicine and Pharmacy “Gr. T. Popa”, 700115 Iasi, Romania; tanasaingrid@yahoo.com (I.-A.V.); demetrasocolov@gmail.com (D.S.); 9Department of Internal Medicine, Faculty of Medicine, University of Medicine and Pharmacy “Gr. T. Popa”, 700115 Iasi, Romania; rodica.radu@umfiasi.ro

**Keywords:** double-J stents, obstetrics, gynecology, side effects

## Abstract

**Background and Objectives**: Double-J stents are urinary catheters that are frequently used in urology. They are now also used in other specialist areas such as obstetrics and gynecology. However, the use of double-J stents is not without side effects. The aim of this review was to highlight the indications and possible adverse effects of the use of these stents in obstetrics and gynecology. **Materials and Methods**: We analyzed works published after 1995 in the PUBMED, SCOPUS, and Web of Science databases related to the use of double-J stents in obstetrics and gynecology, as well as reported adverse events. We carried out a narrative review of the available literature on this topic. **Results**: We identified 69 relevant publications that we included in the review. In obstetrics, indications include the treatment of gestational hydronephrosis, some urological conditions during pregnancy, such as obstructive urinary calculi, with or without superinfection, or intraoperative use for cesarean section or hysterectomy after cesarean section, to protect from, or to solve, ureteral lesions. In gynecology, they are used preoperatively or intraoperatively to protect the ureter during gynecological operations in the pelvic area or postoperatively to repair some ureteral injuries. They are also indicated for ureteral obstructions that occur after pelvic radiotherapy for gynecological neoplasms. Complications associated with the use of double-J stents include more frequent urinary tract infections, lower urinary tract symptoms, calcifications and misplacements. **Conclusions**: Double-J stents are widely used in obstetrics and gynecology and are characterized by good efficiency and safety, although some side effects may occur (lower urinary tract symptoms, hematuria, complications in birth outcomes), which do not limit their use. Summary of evidence: In this review, we analyzed the indications and complications of double-J ureteral stenting in obstetric and gynecologic patients. We found that the procedure is safe, both in the treatment of ureteral obstruction and in the resolution of postoperative complications. No serious complications of ureteral stenting have been noted that would constitute a contraindication to its use. Future prospective studies in large patient cohorts are necessary to validate our data.

## 1. Introduction

Double-J ureteral stents are urinary catheters that are frequently used in urology for a variety of operations. They are used to drain urine from a potentially obstructed ureter and to stabilize the ureter after surgery [1]. These catheters are also used in other specialties [2,3,4,5] when performing abdominal surgeries that may involve the urinary system. Double-J stents are now widely used in obstetrics and gynecology. Many conditions benefit from the insertion of these stents, so obstetrics and gynecology physicians are relatively frequently confronted with patients with indwelling double-J stents or in whom there is an indication for the insertion of a double-J stent to resolve some gynecologic complications and conditions that occur in obstetrics or secondary to pregnancy.

However, there is controversy in the literature about the efficacy and safety of the use of double-J stents, including in obstetrics and gynecology. Some studies report numerous side effects, some of them serious [6,7], which would contraindicate or drastically limit their use [8].

In recent years, the number of articles reporting on the use of double-J stents in obstetrics and gynecology has increased, but there is no review in the existing literature that summarizes all this information.

We therefore reviewed the literature to determine the current indications for double-J stent use and reported adverse events in obstetrics and gynecology to help gynecologists better understand when double-J stents should be used and the risks associated with them and also to help urologists evaluate the indications for double-J stent placement when faced with a urologic complication in obstetrics and gynecology.

## 2. Materials and Methods

We searched the PUBMED, SCOPUS, and Web of Science databases for relevant articles in English and French published after 1995. The search was conducted between 1 September and 30 September 2024, using the following keywords: “Double-J stent”, “pregnancy”, “obstetrics”, “gynecology”, “complications of double-J stent in pregnancy”, “complications of double-J stent in gynecology”, and “complications of double-J stent in obstetrics”. All articles on double-J stents in obstetrics and gynecology were analyzed, including systematic reviews, meta-analyses, cohort and case–control studies, and clinical case reports. The literature search yielded 252 publications, which were screened for eligibility by 3 researchers. We excluded duplicate articles, articles without an abstract, and those with an abstract in a language other than English or French. We also excluded articles that were not relevant or not related to our topic. We finally identified 69 full-text research articles on our topic, which we included in the analysis.

In total, 22 articles dealt with the use of double-J stents in obstetrics; 15 dealt with gestational hydronephrosis and ureteral stone obstruction in pregnancy, and 8 addressed pregnant women with placenta accreta. A total of 21 articles dealt with the use of double-J stents in gynecology; 2 of them dealt with the use of stents in gynecological pelvic neoplasms, 2 articles addressed the use of double-J stents in retroperitoneal fibrosis as a result of radiotherapy for female genital neoplasms, 6 articles dealt with ureteral stenting before and during surgery for female genital neoplasms, and 11 articles dealt with ureteral lesions discovered after hysterectomy for genital neoplasms. A total of 41 articles dealt with the occurrence of complications associated with the use of double-J stents in obstetrics and gynecology, including 7 articles on the calcification of double-J stents, 7 articles on the risk of infectious complications associated with the use of double-J stents, 10 articles on lower urinary tract symptoms, 7 articles on uretero-iliac fistula in patients with indwelling double-J stents, 3 articles on the misplacement and/or migration of double-J stents, 1 article on the rupture of the stent, 3 articles on the incorrect insertion of the catheter, and 4 articles on the consequences of pregnancy resulting from the use of double-J stents. Of these articles, 10 articles dealing with both indications and complications of double-J stents in obstetrics and/or gynecology were cited between 2 and 4 times in this review.

## 3. Results

### 3.1. Utilization of Double-J Stents in Obstetrics

A.Gestational hydronephrosis and ureteral stone obstruction in pregnancy

Gestational hydronephrosis occurs in 0.5% of all pregnancies [9], with most being asymptomatic [10]. If the degree of hydronephrosis is greater, then pain [11] or superinfection [12] may occur, necessitating the insertion of a double-J stent for pain relief or the drainage of the infected hydronephrosis [13,14], even if the hydronephrosis is associated with the presence of a perirenal urinoma [15]. The insertion of the double-J stent is usually indicated after conservative medical treatment has been attempted. In a study by Tsai et al. comparing 25 patients with conservative treatment and 25 patients with double-J stenting for the treatment of hydronephrosis, they found that double-J stenting had better efficacy in successful treatment compared with conservative treatment (100% vs. 80%, *p* = 0.018). However, due to complications in four cases (16%) (discomfort with the stent and flank pain), the authors concluded that conservative treatment should be attempted first, and only if double-J stent insertion has failed. It should be noted that the double-J stents remained in place for an average of 4.5 ± 1.3 months [9]. Ureteral stents have been used as an alternative to percutaneous nephrostomy [16] or conservative treatment [17]. In a study of 84 patients, Şimşir et al. compared the success rate of double-J stenting with that of percutaneous nephrostomy in the treatment of symptomatic gestational hydronephrosis. They found that secondary intervention was required more frequently in the double-J group (34.7% vs. 15.7%, *p* = 0.0018) due to higher degrees of hydronephrosis and pain attributable to the dislocation or encrustation of the double-J. In addition, the time before a secondary intervention was needed was shorter in the double-J group (18 days vs. 33 days, *p* = 0.0025 [16]). Double-J stents are inserted under local anesthesia [12] and can remain in place for an average of 9 weeks [17] until after delivery [17,18] and can be replaced 4–5 weeks after delivery [5].

Renoureteral lithiasis over 8 mm in pregnancy can cause hydronephrosis with renal colic [19] and superinfection, a high degree of hydronephrosis, fever, leukocytosis, elevated creatinine, and CRP [13]. The indication for the insertion of the double-J stent in lithiasis would be ureteral stones. When inserting the double-J stent, the chances of eliminating ureteral stones while wearing the stent are low [20]. Therefore, several studies have compared the insertion of double-J stents with retrograde ureteroscopy in pregnancy. Some show the superiority of ureteroscopy [21], others recommend the insertion of the double-J stent as a first step [20] or assume that both have a similar efficiency [22]. However, a double-J stent has also been inserted postoperatively during ureteroscopy, but it had already been suppressed during pregnancy [21].

Most studies consider double-J stenting to be a simple, efficient, and safe procedure with no significant side effects [12,14,19,22]. However, Tan et al. reported a success rate of only 83.3% in 5 of the 30 patients studied in whom ureteral stenting was attempted but stent placement failed, necessitating percutaneous nephrostomy [22].

B.Placenta accreta spectrum

The presence of a placenta accreta presents obstetricians with numerous difficulties and necessitates a cesarean section, which may later require a hysterectomy for hemostasis [23,24]. These procedures can result in bladder ruptures that extend close to the ureteral orifices and require the insertion of double-J stents to protect them [25]. Although the most common injuries during these operations are bladder injuries [26], partial or complete ureteral injuries have also been described in the literature [27,28]. To avoid these, some authors have used double-J ureteral stents preoperatively and report either a reduction in the rate of ureteral injury [24] or no reduction [29]. In a comparative study by Crocetto et al., in which they analyzed 24 (54.5%) patients with placenta accreta and ureteral stents placed prior to hysterectomy and 20 (45.5%) patients with placenta accreta and no preoperatively placed double-J stent, they found the same rate of intraoperative bladder injury (25% vs. 10%, *p* = 0.21) without detecting ureteral injuries, calling into question the efficacy of preoperative ureteral stenting in preventing bladder and ureteral injuries at the time of cesarean hysterectomy [29]. For ureteral lesions detected intraoperatively during cesarean section or hysterectomy, the insertion of a double-J stent is the treatment of choice, although some authors still prefer percutaneous nephrostomy [27]. In the case of total pelvic ureteral lesion, ureterovesical reimplantation with the insertion of a double-J stent is indicated [27]. Ureteral lesions are occasionally discovered postoperatively, in which case the double-J stent can also be inserted anterograde, if retrograde, transurethral insertion has failed [30]. In a case of a posthysterectomy hematoma with the extrinsic compression of the ureter, a double-J stent was inserted for 3 months until the pelvic hematoma regressed [26].

### 3.2. Utilization of Double-J Stents in Gynecology

A.Genital neoplasms of the pelvis with secondary ureterohydronephrosis

Obstructive uropathy is a common finding in advanced cervix carcinoma [31]. In these cases, a double-J ureteral stent is recommended to facilitate urinary passage. In a retrospective study, Warli et al. examined 88 patients with advanced cervical cancer in whom double-J stent insertion was successful in 43 (48.9%) cases and unsuccessful in 45 (51.1%) cases. They found that the predictors of successful double-J stent insertion were a lower level of hydronephrosis (OR: 18.203, *p* = 0.001), urea (OR: 4.207, *p* = 0.037), and creatinine (OR, 6.923, *p* = 0.004) and a lower stage of cervical cancer (OR: 4.125, *p* = 0.022) [31]. In malignant ureteral obstruction mainly due to cervical cancer [31], double-J stents can be used to avoid percutaneous nephrostomy, but they need to be replaced frequently [32], so the use of metal stents is preferred, as they require fewer replacements and have a better cost–benefit ratio [32].

B.Retroperitoneal fibrosis with unilateral or bilateral ureterohydronephrosis secondary to radiotherapy for pelvic genital neoplasms or breast neoplasms

Retroperitoneal fibrosis occurring after radiotherapy can cause ureteral obstruction and secondary chronic renal failure. In these cases, the first treatment option is ureteral stenting with double-J stents [33]. They can remain in place over the long term, and ureterolysis can be performed later [33]. Metal 6 Fr stents can also be used successfully as an alternative to 6–7 Fr polyurethane double-J stents [34].

C.Ureteral stenting before interventions for endometriosis and uterine neoplasms

Cases with endometriosis of the bladder or around the pelvic ureters have been reported in the literature [4,35,36,37] in which the authors inserted bilateral double-J ureteral stents via cystoscopy prior to laparoscopic or robotic surgery [38]. Piryev et al. published a comparative study of three groups of patients undergoing deep endometriosis surgery. In the first group, the double-J stents were left in place for 2 weeks postoperatively; in group 2, the double-J stents were removed immediately at the end of the procedure, and in group 3, the double-J stents were not inserted [37]. As expected, they found that the groups with double-J stents had a higher rate of urinary tract infections, especially in group 1. Surprisingly, however, the rate of intraoperative ureteral injury was the same in all three groups, suggesting that the insertion of double-J stents offered no advantage in protecting against intraoperative ureteral injury. Ureteral stents were also inserted when only the bladder dome was removed, without performing a hysterectomy [39]. A case has also been reported in which fluorescent stents were used preoperatively during hysterectomy for uterine cancer [40].

D.Stenting during operations for gynecological pathologies

Total hysterectomies with pelvic lymph dissection, which are indicated for uterine neoplasms [41] or uterine fibroids [42], carry the risk of ureteral injury, regardless of whether the procedure is performed open or laparoscopically [43,44]. In a study by Yan et al., of 117 patients treated for cervical cancer who underwent laparoscopic radical hysterectomy and pelvic lymphadenectomy, a case of postoperative ureteral fistula requiring the cystoscopic insertion of a double-J stent that was left in place for 8 weeks was described [43]. To avoid accidental ureteral injury during hysterectomy, double-J stents are sometimes inserted preoperatively to protect the ureters, and fluorescent catheters are preferred [45]. In most cases, ureteral lesions are detected intraoperatively [41,46], necessitating ureterovesical reimplantation with double-J stents [41] or end-to-end ureteral reanastomosis using double-J stents [47]. In a report by Han et al. on 12 cases of ureteral transection during gynecologic laparoscopy, ureteroureterostomy was performed in eleven cases and laparoscopic ureteroneocystostomy with a double-J stent was performed in one case. These stents were left in place for a period of 73 days [44]. Other studies report that double-J stents inserted during laparoscopic repair remain in place for between 6 and 16 weeks [41]. In other cases, ureteral lesions are discovered late postoperatively, usually in the form of ureterovaginal fistulas [48]. In these cases, a double-J stent is placed either by cystoscopy [49] or by ureteroscopy [48], which is the recommended initial treatment [49]. To improve the cure rate with ureteral stenting, Deng et al. recommend prior ureteroscopy to visualize the ureteral lesion and decide whether double-J stenting with 4.7 Fr/26 cm should be omitted and ureteroneocystostomy should be performed from the beginning if the rupture is more than half of the ureteral circumference [48]. A study by Chung et al. found that the most common ureteral injuries occurred after gynecologic procedures, mostly laparoscopic. The diagnosis was made a median of 17.4 days (range 0–68 days) after surgery, and the median time between injury and stenting was 19.4 days (0–63 days). Successful retrograde stenting was achieved in 18 cases. In one case where double-J stenting failed, percutaneous nephrostomy was required. Ureteral strictures were reported after a median follow-up of 9.7 months (range, 1 month to 2 years), demonstrating that in some cases, double-J stenting failed to prevent this complication [49]. The most common gynecologic surgeries in which ureteral injuries occur are hysterectomies, but they can also occur in cesarean sections and cesarean hysterectomies where the retention time of the double-J stents is 6–8 weeks [46]. The success rate after double-J stenting is higher after obstetric and gynecologic procedures than after urologic and abdominal procedures [46]. A case of spontaneous ureteral rupture after chemotherapy for cervical cancer has also been reported [50], with secondary ureterohydronephrosis, in which a double-J stent was inserted, leading to the conclusion that in cases of ureterohydronephrosis before chemotherapy, it would be advisable to insert a double-J stent in advance to avoid such incidents. In addition, a case of pyosalpinx with ureteral compression and secondary ureterohydronephrosis was reported in which a double-J ureteral stent was inserted for desobstruction [51]. The indications for ureteral stenting in obstetrics and gynecology are summarized in Table 1.

### 3.3. The Complications of Using Double-J Stents in Obstetrics and Gynecology

A.Calcifications of the double-J stents

Despite the wide use of double-J stents in obstetrics and gynecology, we found only six articles [22,51,52,53,54,55] reporting calcifications of double-J stents. In one pregnant woman, the calcification of the catheter on both the proximal and distal loops was reported only one week after catheter insertion, which consisted of calcium phosphate and required endoscopic bladder lithotripsy for removal [52]. In a 40-year-old female patient, calcifications occurred in both the proximal and distal loops after 4 years [56]. Among 30 pregnant women who underwent double-J stents for intractable flank pain or flank pain with sepsis, calcifications occurred in 3 of them (10%) when the stent was removed [53]. However, in a series of over 50 cases, the incrustation rate was only 3.84% [55]. Due to the incrustation, the double-J stent can hardly ever be removed [22].

B.The risk of infection in double-J stent carriers

As the risk of infection in double-J stent wearers is recognized in the literature [57,58], it was to be expected that studies on this topic would be highlighted. In a study, Chen et al., who investigated the incidence of acute pyelonephritis in double-J stent carriers compared to non-carriers, found a higher incidence of pyelonephritis in pregnant patients with double-J stents compared to pregnant patients without double-J stents (9% vs. 4%). In general, the risk of pyelonephritis is higher in pregnant women with double-J stents than in women without double-J stents [58], and pregnancy has also been shown to be a risk factor (*p* < 0.01) for lower urinary tract infection in patients with double-J stents [6]. Reflux pyelonephritis in patients with double-J stents can have an unfavorable clinical course and lead to urosepsis [18]. There is also an increased risk of bacterial colonization, in both the proximal and distal loop [6]. The occurrence of a psoas abscess requiring drainage was reported in a pregnant patient who had a double-J stent inserted due to infected hydronephrosis [59]. An infection rate of 20.8% was reported in pregnant patients who had double-J stents [60], although no urinary tract infections were reported in other studies, even when the catheters were worn for more than 2 months [17]. The risk of pyelonephritis has also been studied in patients undergoing double-J stent insertion for malignant obstruction, with no evidence of increased risk [61]. In a recent study, Anton et al. showed that pregnant women with double-J stents have an increased risk of multidrug-resistant urinary tract infections compared to pregnant women without double-J stents (12.85% vs. 0%, *p* = 0.01) [62].

C.The presence of lower urinary tract symptoms (LUTSs) and hematuria

The presence of lower urinary tract symptoms and hematuria in double-J stent wearers has also been reported in obstetric and gynecologic patients. For example, in a study by Lim et al. in patients with cervical cancer, endometrial cancer, and retroperitoneal fibrosis following radiotherapy for cervical cancer wearing a polyurethane double-J ureteral stent (Cook, IL, USA, 6 Fr, 24–26 cm), the occurrence of pollakiuria, urinary retention, urinary urge, intermittency, hesitancy, a feeling of incomplete bladder emptying, and interrupted urinary stream was reported. These symptoms improved over time but significantly reduced quality of life after 9 months [7]. However, the presence of voiding symptoms and discomfort did not require double-J stent suppression [12]. Also, in pregnant women with ureteral lithiasis in whom double-J stents were inserted, stent tolerability was poor due to low urinary voiding phenomena, so the authors recommended ureteroscopy instead of double-J stent insertion [63]. Similarly, Çeçen et al. recommended conservative medical treatment instead of double-J stent insertion for hydronephrosis in pregnancy. They were unable to demonstrate a reduction in complication rate, postpartum pain score, and permanent hydronephrosis (*p* > 0.05) in patients with double-J stents compared to patients without double-J stents, even though they reported no serious complications during or after double-J stent insertion [17]. A 29.2% incidence of hematuria and a 20.8% incidence of lower urinary tract phenomena were also reported in pregnant women [60]. Another study reported a 17% incidence of hematuria and LUTSs after double-J insertion during pregnancy [53]. Even though the complication rate after double-J insertion is up to 45%, including LUTSs, these complications were minor (classified according to Clavien–Dindo 1, 2, and 3) [20]. However, some authors reported a low incidence of bladder irritation symptoms and hematuria [24]. Regarding the risks of the preoperative installation of double-J stents prior to gynecologic surgery, it was found that patients with preoperative stents operated on for malignant disease had a higher risk of urinary tract infection, re-stenting, hydronephrosis, and readmission compared to patients operated on for benign disease [64]. As an alternative to double-J stenting, Liang et al. suggested the use of ureteral catheters instead of double-J stents in patients with placenta accreta spectrum who underwent hysterectomy and demonstrated a lower risk of bladder irritation (aOR, 0.186; 95% CI, 0.057–0.605, *p* = 0.005), hematuria (aOR, 0.011; 95% CI, 0.001–0.136, *p* < 0.001), and lower back pain (aOR, 0.075; 95% CI, 0.022–0.261, *p* < 0.001) [24].

D.Ureteroiliac fistula in chronic double-J stent wearers

Chronic carriers of double-J stents, especially those with retroperitoneal fibrosis after radiotherapy for pelvic genital neoplasms [8,65], may develop compression necrosis leading to the development of a fistula between the ureter and the iliac artery. Patients usually present with intermittent hematuria [66], but they can also present with massive hematuria and hemorrhagic shock, with a mortality of 7–23% [67]. Usually, the fistula does not occur until months or even years after the insertion of the double-J stent [68,69]. Treatment usually consists of the insertion of a stent into the iliac artery or open surgery [66,70]. Sometimes the fistula can occur bilaterally [68], at different intervals, and sometimes it can even occur after 2 days [69].

E.Misplacement, migration, or clogging of the double-J stent

Double-J stent migration has been reported in 3 pregnant women out of 30, in whom they were implanted due to hydronephrosis with or without ureteral lithiasis and required repositioning [53]. In one pregnant woman, a double-J stent that had migrated into the cardiovascular system was objectified, and clinicians waited until after delivery to remove it. Sometimes, the double-J stent does not achieve efficient urinary drainage, and hydronephrosis persists or worsens, eventually requiring nephrectomy [27] or more frequent stent replacement compared to percutaneous nephrostomy [16].

F.The impossibility of inserting the double-J stent or incorrect insertion

Double-J stents are placed preoperatively by cystoscopy or intraoperatively during open or laparoscopic gynecologic procedures. Cases have been reported in which the double-J stent could not be inserted intraoperatively, so a percutaneous nephrostomy had to be inserted [71]. In one case of retroperitoneal fibrosis following radiotherapy for uterine cancer, the catheter inserted via cystoscopy was mistakenly inserted into the vena cava [72]. In a series of 131 cases in which ureteral stenting was inserted, the pooled surgical success rate was 97% for the insertion of a double-J stent [20].

G.Effects on birth and fetus

In the case of urosepsis in pregnant women due to obstructive pyelonephritis, the insertion of the double-J stent did not prevent fetal suffering in the event of septic shock [18]. No maternal complications or postpartum fetal distress were reported in pregnant women who underwent double-J stenting for renal colic [73]. A higher rate of cesarean section and a higher rate of postpartum urinary tract infection and an increased risk of the premature rupture of membranes were reported in pregnant women with urosepsis in whom a double-J stent was inserted and worn until delivery (aOR, 5.59; 95% CI, 2.02–15.40, *p* < 0.001) as well as a higher rate of preterm birth (aOR, 2.47; 95% CI, 1.15–5.33, *p* = 0.02), but without an increase in the incidence of pre-eclampsia (aOR, 2.07; 95% CI, 0.63–6.65, *p* = 0.22) or intrauterine growth restriction (aOR, 1.23; 95% CI, 0.48–3.14, *p* = 0.66) [55,74]. A systematic review by Jin et al. examined the results of treatment with internal ureteral stents or ureteroscopy in pregnant women with urolithiasis; it was found that in the group with stents, the rate of normal fertility outcomes was 99% and the rate of preterm births and abortions was less than 1%, similar to that in pregnant women who underwent ureteroscopy and did not have ureteral stents [20].

A summary of complications associated with double-J stents being used in obstetrics and gynecology can be found in Table 2.

## 4. Strengths and Limitations

The major strength of our study lies in the compilation of information on the use of double-J stents in obstetrics and gynecology, indications and complications, and data that have been reported separately in the literature to date and generally on small patient groups. The present review is an up-to-date retrospective on the use of double-J stents in obstetrics and gynecology, which will be of interest to physicians in this specialty but also to urologists who have to intervene in such cases. Our study has several limitations. The studies included in the review are mostly retrospective studies or even clinical case reports, as these cases are rare, especially when it comes to obstetric and gynecological complications. Reports on the use of double-J stents often do not specify the type of catheter used, the length and thickness, and the duration of stenting implantation, which influences the reporting of their efficacy and, in particular, of their possible complications. Some of the studies included in this analysis reported the use of double-J stents without specifying their efficacy compared to patients in whom no double-J stents were used in the same context. There was a large heterogeneity of patients included in the study in terms of the number of patients, study period, and follow-up time.

## 5. Conclusions

Ureteral stenting has numerous indications in obstetrics and gynecology. It is a safe and effective method that is sometimes indispensable in resolving certain gynecological complications. The complication rate in double-J stents wearers is low, with a predominance of lower urinary tract symptoms, but this does not argue against the widespread use of double-J stents.

## 6. Future Directions

In the future, prospective studies on larger patient groups and also on particular patient groups (pregnant patients with recurrent urolithiasis) will be necessary to validate our results. In addition, the influence of catheter types (materials, thickness), stenting duration and the presence of biofilms on the efficiency of their use in obstetrics and gynecology needs to be analyzed.

## Figures and Tables

**Table 1 jcm-13-07649-t001:** Indications of double-J stents in obstetrics and gynecology.

Utilization of double-J stents in obstetrics	A. Gestational hydronephrosis and ureteral stone obstruction in pregnancy	gestational hydronephrosis
		ureteral stone obstruction in pregnancy
		obstructive pyelonephritis in pregnancy
	B. Placenta accreta spectrum	protection of the ureters before cesarean sections in placenta percreta
		resolution of some ureteral injuries during cesarean section or haemostatic hysterectomy in patients with placenta percreta
		intraoperatively, to protect the ureter, during hysterectomy operations, for accidental bladder injuries, in patients with placenta percreta
Utilization of double-J stents in gynecology	in pelvic genital neoplasia, especially in advanced uterine neoplasia, with secondary ureterohydronephrosis	
	patients with retroperitoneal fibrosis and secondary ureterohydronephrosis after radiotherapy for pelvic genital neoplasia	
	preoperatively, before radical hysterectomy for pelvic genital neoplasms or endometriosis	
	intraoperatively, in ureteral lesions or in hysterectomy operations for genital neoplasia or other benign pathologies	

**Table 2 jcm-13-07649-t002:** Complications associated with double-J stents in obstetrics and gynecology.

1.	Calcifications of double-J stents
2.	Urinary tract infections, reflux pyelonephritis, urosepsis
3.	Lower urinary tract symptoms (LUTSs), gross hematuria
4.	Ureteroiliac fistula
5.	Misplacement, migration, or clogging of the double-J stent
6.	The impossibility of inserting the double-J stent or incorrect insertion
7.	Complications in birth outcomes

## Data Availability

Not applicable.

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
