# Peer review of "Double-J Ureteral Stenting in Obstetrics and Gynecology: Pivotal or Problematic?"

_jcm, 2024, doi:10.3390/jcm13247649_

Round 1
Reviewer 1 Report
Comments and Suggestions for Authors
The manuscript provides a narrative review of the use of double-J stents in obstetrics and gynecology, discussing indications, complications, and outcomes. The structure is logical, with detailed sections on clinical applications, challenges, and future directions. However, some areas could benefit from refinement in clarity, consistency, and depth.
Major Comments:
1) The manuscript identifies itself as a narrative review but references adherence to the PRISMA extension for scoping reviews, leading to some confusion about the review type. A narrative review and a scoping review differ significantly in purpose, methodology, and outcomes. If the manuscript is intended to be a narrative review, explicitly state this in the methodology and remove references to scoping review frameworks such as PRISMA. If the manuscript aims to be a scoping review, additional methodological details must be provided, including systematic inclusion and exclusion criteria, a detailed search strategy, and a comprehensive charting of results.
2) The review novelty is unclear. While it states that no prior review summarizes this information, comparing this manuscript's scope to existing literature would strengthen its relevance.
3) Consider incorporating a table that ranks studies by quality or highlights their limitations (e.g., small sample size, retrospective nature).
4) The manuscript effectively presents the benefits of double-J stents, but the risks and complications deserve more detailed exploration. For example, discuss the management of stent-related complications, including LUTS and encrustations, with evidence-based recommendations and explore the alternatives to stents, such as percutaneous nephrostomy or ureteroscopy, in greater detail for comparative analysis.
5) While the authors propose future studies, more specific suggestions would enhance this section. For instance, identifying particular patient subgroups (e.g., pregnant patients with recurrent urolithiasis) or addressing optimal stenting duration would provide clear guidance.
Minor Comments:
1) Standardize terminology throughout the manuscript. For example, "double-J stent" and "double J catheter" are used interchangeably.
2) Ensure consistency in reporting metrics (e.g., survival rates, complication rates) for easier comparison across studies.
3) Table 2 in its current form is not structured as a proper table but rather as a bulleted list of complications associated with double-J stents.
4) A significant portion of the references (19 out of 74, approximately 26%) are over 10 years old.
Author Response
The manuscript provides a narrative review of the use of double-J stents in obstetrics and gynecology, discussing indications, complications, and outcomes. The structure is logical, with detailed sections on clinical applications, challenges, and future directions. However, some areas could benefit from refinement in clarity, consistency, and depth.
Major Comments:
- The manuscript identifies itself as a narrative review but references adherence to the PRISMA extension for scoping reviews, leading to some confusion about the review type. A narrative reviewand a scoping review differ significantly in purpose, methodology, and outcomes. If the manuscript is intended to be a narrative review, explicitly state this in the methodology and remove references to scoping review frameworks such as PRISMA. If the manuscript aims to be a scoping review, additional methodological details must be provided, including systematic inclusion and exclusion criteria, a detailed search strategy, and a comprehensive charting of results.
- Thank you for your comments. We removed the references to scoping review frameworks.
- The review novelty is unclear. While it states that no prior review summarizes this information, comparing this manuscript's scope to existing literature would strengthen its relevance.
- In the existing literature there is no review to summarise the indications and complications of double-J stenting in obstretrics and gynecology. So, we think that this review adds new informations to the topic useful for both urologists and gynecologists.
- Consider incorporating a table that ranks studies by quality or highlights their limitations (e.g., small sample size, retrospective nature).
- Our study is a narrative review and it’s beyond our purpose to rank the studies. Anyway we have specified in our paper that the vast majority of articles are retrospective and most of them with relatively small sample size.
- The manuscript effectively presents the benefits of double-J stents, but the risks and complications deserve more detailed exploration. For example, discuss the management of stent-related complications, including LUTS and encrustations, with evidence-based recommendations and explore the alternatives to stents, such as percutaneous nephrostomy or ureteroscopy, in greater detail for comparative analysis.
- Initially we wanted to write about the management of these complications BUT haven`t found data in the literature about this on the particular group of patients. Regarding the alternative to ureteral stenting, we have already discussed percutaneous nephrostomy and ureteroscopy, and we inserted all the data found in the articles about this topic.
- While the authors propose future studies, more specific suggestions would enhance this section. For instance, identifying particular patient subgroups (e.g., pregnant patients with recurrent urolithiasis) or addressing optimal stenting duration would provide clear guidance.
- Thank you, we have modified according to your recommendations.
Minor Comments:
- Standardize terminology throughout the manuscript. For example, "double-J stent" and "double J catheter" are used interchangeably.
- We have modified accordingly: double- J catheter with double-J stent.
2) Ensure consistency in reporting metrics (e.g., survival rates, complication rates) for easier comparison across studies.
3) Table 2 in its current form is not structured as a proper table but rather as a bulleted list of complications associated with double-J stents.
- Yes, indeed. It’s rather a list of complications but is inserted in table form.
4) A significant portion of the references (19 out of 74, approximately 26%) are over 10 years old.
- It is true but, there are references reporting several complications and indications and there have been no recent studies on this particular group of patients. This is why we consider this articles relevant to our topic.

Reviewer 2 Report
Comments and Suggestions for Authors
The authors aimed to highlight the indications and possible adverse effects of the use of these stents in obstetrics and gynecology.
They analyzed the papers from 1995 in the PUBMED, SCOPUS, and Web of Science databases related to the use of double-J stents in obstetrics and gynecology, as well as reported adverse events, using the PICO method. Why did the authors use 1995 as a threshold?
Moreover, they relied on 69 papers. In the review, they included also the pregnancy. In my opinion, pregnancy should be addressed separately. Indeed, the pregnancy may induce modifications of pelvic organs and urinary tract that should be better acknowledged and are worthy to be discussed separately (PMID 38999471).
Table 1 should be re-edited in a nicer format.
Author Response
The authors aimed to highlight the indications and possible adverse effects of the use of these stents in obstetrics and gynecology.
They analyzed the papers from 1995 in the PUBMED, SCOPUS, and Web of Science databases related to the use of double-J stents in obstetrics and gynecology, as well as reported adverse events, using the PICO method. Why did the authors use 1995 as a threshold?
- The utilization of double-J stents started roughly 30years ago. That`s why we included in our analysis articles on this period of time and we didn’t searched for older studies, because was little chance to find relevant articles for our topic.
Moreover, they relied on 69 papers. In the review, they included also the pregnancy. In my opinion, pregnancy should be addressed separately. Indeed, the pregnancy may induce modifications of pelvic organs and urinary tract that should be better acknowledged and are worthy to be discussed separately (PMID 38999471).
- As we stated in the title of the review we took in consideration also obstetric patients. We haven`t found many articles about the indications and complications of double-J stenting during pregnancy so to study separately this particular type of patients.
Table 1 should be re-edited in a nicer format.
- We have modified Table 1.

Reviewer 3 Report
Comments and Suggestions for Authors
This review manuscript has scientific merit that might benefit readers, but some minor revisions still need to be included. Therefore, we recommend that the authors make the following modifications:
1. Line 28-29:
“The use of double-J stents in obstetrics and gynecology, as well as reported adverse events, using the PICO method.”
The authors are suggested to add the full abbreviation of PICO in the abstract for a clearer understanding for the readers.
2. Line 39-40 (Abstract):
"The double-J ureteral catheters are widely used in obstetrics and gynecology and are characterized by good efficiency and safety, although some side effects may occur, which does not limit their use."
The authors are suggested to kindly add a few main side effects that they reviewed through 69 publications related to Double-J ureteral stenting in obstetrics and gynecology in the conclusion section of the abstract.
3. Line 61-63:
“However, there is controversy in the literature about the efficacy and safety of the use of double-J catheters, including in obstetrics and gynecology. Some studies report numerous side effects, some of them serious, which would contraindicate or drastically limit their use.”
The authors are suggested to add the details about the controversy in the literature about the efficacy and safety in a table format as key notes, along with some key findings from the previous studies reporting numerous side effects.
Table no. 1: Efficacy and safety of the use of double-J catheters, including in obstetrics and gynecology
|
Efficacy and safety & Contradictory effects of Double-J catheters |
|
|
Obstetrics |
Gynecology |
|
|
|
|
|
|
4. Line 27:
“We analyzed publications published after 1995 in PUBMED, SCOPUS.”
Line 74:
“We searched the PUBMED, SCOPUS, and Web of Science databases for relevant articles in English and French published after 1990.”
In the abstract, the authors mention the literature survey starts from 1995, but in Section 2 (Materials and Methods), the authors mention 1990. We suggest the authors review the literature review again to clarify this discrepancy.
5. Line 50-73:
Section 1. Introduction
The authors are suggested to make a schematic diagram that shows Double-J ureteral stenting in obstetrics and gynecology.
Figure 1: Schematic diagram of Double-J ureteral stenting in obstetrics and gynecology and main chief complications.
This figure will help to illustrate the main course of the review and make it more comprehensive while addressing the key issues of the narrative review manuscript.
6. Line 86-87:
“The article selection flowchart is shown in Figure 1.”
The authors are suggested to replace Figure 1. with the above-mentioned suggested Figure 1: Schematic diagram of Double-J ureteral stenting in obstetrics and gynecology and main chief complications.
The original Figure 1 can be added to the supplementary data. This will make the manuscript more constructive.

Author Response
- Line 28-29:
“The use of double-J stents in obstetrics and gynecology, as well as reported adverse events, using the PICO method.”
The authors are suggested to add the full abbreviation of PICO in the abstract for a clearer understanding for the readers.
- We have added.
- Line 39-40 (Abstract):
"The double-J ureteral catheters are widely used in obstetrics and gynecology and are characterized by good efficiency and safety, although some side effects may occur, which does not limit their use."
The authors are suggested to kindly add a few main side effects that they reviewed through 69 publications related to Double-J ureteral stenting in obstetrics and gynecology in the conclusion section of the abstract.
- We have added as suggested
- Line 61-63:
“However, there is controversy in the literature about the efficacy and safety of the use of double-J catheters, including in obstetrics and gynecology. Some studies report numerous side effects, some of them serious, which would contraindicate or drastically limit their use.”
The authors are suggested to add the details about the controversy in the literature about the efficacy and safety in a table format as key notes, along with some key findings from the previous studies reporting numerous side effects.
Table no. 1:Efficacy and safety of the use of double-J catheters, including in obstetrics and gynecology
|
Efficacy and safety & Contradictory effects of Double-J catheters |
|
|
Obstetrics |
Gynecology |
|
|
|
|
|
|
- Line 27:
“We analyzed publications published after 1995 in PUBMED, SCOPUS.”
Line 74:
“We searched the PUBMED, SCOPUS, and Web of Science databases for relevant articles in English and French published after 1990.”
In the abstract, the authors mention the literature survey starts from 1995, but in Section 2 (Materials and Methods), the authors mention 1990. We suggest the authors review the literature review again to clarify this discrepancy.
- We have corrected – 1995 instead of 1990.
- Line 50-73:
Section 1. Introduction
The authors are suggested to make a schematic diagram that shows Double-J ureteral stenting in obstetrics and gynecology.
Figure 1:Schematic diagram of Double-J ureteral stenting in obstetrics and gynecology and main chief complications.
This figure will help to illustrate the main course of the review and make it more comprehensive while addressing the key issues of the narrative review manuscript.
- We inserted in table 1 the diagram of the indications of double-J stenting in obstetrics and gynecology and in table 2 we listed the complications. So, we don`t think that is necessary to summarize this tables.
- Line 86-87:
“The article selection flowchart is shown in Figure 1.”
The authors are suggested to replace Figure 1. with the above-mentioned suggested Figure 1: Schematic diagram of Double-J ureteral stenting in obstetrics and gynecology and main chief complications.
The original Figure 1 can be added to the supplementary data. This will make the manuscript more constructive.
- Our review has 2 main parts: the indications and the complications of the double-J stenting. So we consider that we must show separately in each part the figures which summarize the data.

Round 2
Reviewer 1 Report
Comments and Suggestions for Authors
The authors have made only superficial revisions in response to my previous comments, addressing issues such as removing references to PRISMA and standardizing the term "double-J catheter" as "double-J stent." I can accept the omission or explanation of several of my comments, BUT two critical issues still require attention.
First, although the manuscript now explicitly identifies itself as a narrative review and removes references to PRISMA, it is clear that the authors still do not fully grasp the distinctions between various review types. To address this, the abstract and the "Materials and Methods" section must avoid using terminology typically associated with systematic or scoping reviews, such as “PICO method” and flowchart depictions, as these are inappropriate for narrative reviews.
Second, if the authors do not agree to transform Table 2 into a simple list, they have to, at the very least, add clear titles to both tables.
Author Response
The authors have made only superficial revisions in response to my previous comments, addressing issues such as removing references to PRISMA and standardizing the term "double-J catheter" as "double-J stent." I can accept the omission or explanation of several of my comments, BUT two critical issues still require attention.
First, although the manuscript now explicitly identifies itself as a narrative review and removes references to PRISMA, it is clear that the authors still do not fully grasp the distinctions between various review types. To address this, the abstract and the "Materials and Methods" section must avoid using terminology typically associated with systematic or scoping reviews, such as “PICO method” and flowchart depictions, as these are inappropriate for narrative reviews.
- Thank you for your comment. We removed the references to PICO method and also the flowchart.
Second, if the authors do not agree to transform Table 2 into a simple list, they have to, at the very least, add clear titles to both tables.
- We added clear titles to both tables.

Reviewer 2 Report
Comments and Suggestions for Authors
No additional comments
Author Response
Thank you.
Reviewer 3 Report
Comments and Suggestions for Authors
Thank you for incorporating all the suggestions. The revised version looks more comprehensive and well-organized.
We wish the authors good luck.
Author Response
Thank you also.